# Investigation of Cutting Force and the Material Removal Mechanism in the Ultrasonic Vibration-Assisted Scratching of 2D-SiCf/SiC Composites

**DOI:** 10.3390/mi14071350

**Published:** 2023-06-30

**Authors:** Hao Lin, Ming Zhou, Haotao Wang, Sutong Bai

**Affiliations:** School of Mechatronics Engineering, Harbin Institute of Technology, Harbin 150001, China; zhouming@hit.edu.cn (M.Z.); 18b308010@stu.hit.edu.cn (H.W.); baisutong1997@126.com (S.B.)

**Keywords:** ultrasonic-assisted scratching, surface formation mechanism, SiCf/SiC composites, surface roughness

## Abstract

Ultrasonic-assisted grinding (UAG) is widely used in the manufacture of hard and brittle materials. However, the process removal mechanism was never elucidated and its potential is yet to be fully exploited. In this paper, the mechanism of material removal is analyzed by ultrasonic-assisted scratching. Three distinct surfaces (S1, S2, and S3) were selected on the basis of the braided and laminated structure of fiber bundles. The ultrasonic-assisted scratching experiment is carried out under different conditions, and the scratching force (SF) of the tested surface will fluctuate periodically. Under the conditions of different feed speeds, depths, and ultrasonic amplitudes, the normal scratching force (S*F_n_*) is greater than the tangential scratching force (S*F_t_*), and the average scratching force on the three surfaces is generally S3 > S2 >S1. Among the three processing parameters, the speed has the most significant influence on the scratching force, while the scratching depth has little influence on the scratching force. Under the same conditions and surface cutting mode, the ultrasonic vibration-assisted scratching force is slightly lower than the conventional scratching force. The scratching force decreases first and then increases with the amplitude of ultrasonic vibration. Because the fiber undergoes a brittle fracture in the ultrasonic-assisted scratching process, the matrix is torn, and the surface residues are discharged in time; therefore, the surface roughness is improved.

## 1. Introduction

In recent years, ceramic matrix composites (CMC), particularly carbon fiber and silicon carbide fiber-reinforced ceramic matrix composites (SiCf/SiC), received extensive attention [1,2]. A SiC ceramic matrix is a composite material with high specific strength and specific stiffness, corrosion resistance, oxidation resistance, high wear resistance, electromagnetic wave absorption, and high resistance, alongside other excellent characteristics [3,4]. However, it also has the disadvantage of being fragile, exhibiting a low fracture toughness [5]. Silicon carbide fiber is compatible with metals, resins, and ceramics, and can be used in heat-resistant, oxidation-resistant, and high-performance composite reinforcement materials. Continuous SiC fiber is used to strengthen the material, allowing the matrix’s excellent properties to be maintained, and its overall toughness improved. Although the cost of the composite materials of silicon carbide fiber (compared with carbon fiber) is greater (because of the organic silicide raw materials produced by the spinning, silicification, or vapor deposition of inorganic fiber with a β-silicon carbide structure), its heat resistance and oxidation properties are better than those of carbon fiber, and the service temperature of silicon carbide fiber can reach up to 1200°. It therefore has excellent development potential in the aerospace, electronic communication, electronic machinery, petrochemical, and biomedical fields. The National Aeronautics and Space Administration (NASA) identified SiCf/SiC as the best material system for developing high-speed civil transport (HSCT) in the research results of their EPM project [6]. Subsequently, this material is used to design and prepare combustion chamber flame tubes, turbine stator cotyledons, wing front segments, thrust chambers, etc. However, this material is also characterized by brittleness and an anisotropic structure and pore characteristics, making it extremely difficult to process and mould into unique shapes and sizes [7,8,9,10,11]. In the process of machining SiCf/SiC ceramic composites, severe tool wear, cracks in the material matrix [12], transverse fractures [13], delamination, and fiber–matrix debonding will occur, resulting in a rough surface and a low processing efficiency, both of which directly affect the performance of the material [14,15]. Due to the material’s anisotropy and hard, brittle properties, experts and engineers used various processing methods to improve its surface machining accuracy and removal efficiency. Due to the defects of delamination and burr in the traditional processing of fiber-reinforced ceramic composites, particular processing technologies, such as pulsed laser machining (PLM), high-pressure water cutting, ultrasonic machining (UM), and electrical discharge machining (EDM), are widely used in the processing of fiber-reinforced ceramic composites [16,17,18,19]. Muttamara et al. successfully drilled microholes into ceramic-based workpieces using the EDM method with an auxiliary electrode [20]. However, SiCf/SiC composites have extremely poor electrical conductivity, so the wear of the tool electrode becomes lager than with the normal machining. Hu et al. used an ultrashort pulse laser to carry out experimental research on the processing of micro through-holes and blind holes in SiCf/SiC composites; they found that it improved the processing quality and accuracy of the wall’s surface, and also that the processed surface will feature a concentration of thermal stress, resulting in multiple cracks [21]. Ultrasonic machining of ceramic composite materials produces a high surface quality, but low efficiency [22,23]. Compared with the normal cutting process, rotary ultrasonic groove machining for ceramic matrix composites significantly reduces cutting force and tool wear [24,25].

In this paper, rotary ultrasonic grinding (RUG) is used to remove ceramic composites. This method can reduce the grinding force, improve the machining accuracy, and reduce the consumption of grinding tools. RUG is effective in improving the surface quality of ceramic composites and reducing tool wear [26,27,28,29,30,31,32,33,34]. However, the removal mechanism of the given material still needs to be fully clarified. Liu et al. studied the influence of different surface grinding forces through Cf/SiC scratching experiments [35]. Yao et al. studied the influence and removal mechanism of the scratching angle and scratching force of SiCf/SiC composites [36]. Ning et al. made a comparison between rotary ultrasonic machining of ceramic composites and conventional machining experiments. Their results show that rotary ultrasonic machining can lead to a larger ductile removal region before the successive brittle fractures and cracks [37]. Therefore, the effect of ultrasonic amplitude on surface roughness can be studied via a single-particle scratching experiment.

SiCf/SiC material is composed of a layer of orthogonal braided fiber bundles superimposed upon the matrix. Different grinding directions produce different machining effects. At present, single-particle scratching is a simple and effective method that is often used to analyze and study the removal mechanism of grinding [38,39,40]. Azarhoushang et al. carried out a comparative test, using ultrasonic-assisted grinding (UAG) and conventional grinding (CG) on C/C-SiC composites; they found that the grinding force was significantly reduced by 20%, and that the surface roughness was reduced by about 30% [41]. Singh R and Khamba JS et al. found that using UM in the high-precision machining of titanium alloys and other alloys can effectively prevent excessive tool wear, reduce the grinding force, and improve the grinding removal rate (GRR) [42].

In this experiment, 2D-SiCf/SiC orthogonal fiber braided stacked ceramic composites were selected as the subject. This material is more rigid and resistant to processing than Cf/SiC. Its fibers can be categorized by three orthogonal directions: transverse fibers, longitudinal fibers, and normal fibers. The effects of grinding in different directions under ultrasonic vibration on the grinding force and machined surface roughness were analyzed. The influence of ultrasonic vibration grinding on three typical grinding surfaces was studied using a Vickers indenter via a single-particle scratching experiment. Different scratch speeds and depth factors were used to study SiCf/SiC composites, obtain the scratching force (S.F.), and deeply analyze images of the composites’ surface morphology. Finally, the effects of ultrasonic vibration scratching and traditional scratching on SiCf/SiC’s material removal rate and surface roughness are compared. This study may provide technical support for improving the machined surface quality of woven fiber-reinforced ceramic matrix composites.

## 2. Material and Rotating Ultrasonic Machining (RUM) Platform

### 2.1. Material Preparation

In the experiment, a SiCf/SiC composite with a two-dimensional orthogonal braided structure was used as the material. Figure 1a shows a scanning electron microscope (SEM) image, and Figure 1b shows an illustration, produced using software, of the braided structure of the SiC fibers. A physical view of the material before processing is shown in Figure 1c. The microstructure of the composite material is shown in Figure 2. The thickness of the orthogonal SiC fiber bundle layer is about 300 μm~500 μm. In this layer, the width of the SiC fiber bundles is about 700 μm, and the arrangement directions are 0° and 90°, respectively. In the fetal layer of SiC fibers, the SiC fibers are composed of SiC fiber bundles 14 μm in diameter, which are irregularly interwoven to form a net-like structure. It can be seen from the image that some SiC fiber bundles are similar to disordered networks, and orderly orthogonal distribution can also be seen in some areas. This is mainly caused by the anisotropic characteristics of the SiCf/SiC material and the internal heterogeneous structure. Therefore, for a single-particle scratching test, irregular fiber network layers should be avoided, and the orthogonal SiC fiber layer should instead be selected as the scratching surface. A simple polishing treatment is needed before the test can proceed. The properties in Table 1 were obtained by testing the material’s physical and mechanical properties and reviewing the resulting data. The sample size of the scratching test material is 15 × 12 × 3 mm. In the polishing process, 40 μm abrasive particles were first selected for grinding (for 2 min); then, 15 μm abrasive particles were chosen for grinding (for 5 min); and finally, 1 μm abrasive particles with suspension were used to grind the surface for 10 min. In this way, the surface can meet the requirements of the test.

### 2.2. Experimental Design and Conditions

The CNC five-axis machining center (Ultrasonic 70-5 linear, DMG MORI, Bielefeld, Germany) was used in the experiment. The USG2000 ultrasonic vibration system equipped with the machining tool had a maximum ultrasonic vibration frequency of 30 kHz. The vibration mode of the tool is such that the spindle moves up and down, and the axial vibration is only one-dimensional (perpendicular to the material surface). A standard Vickers diamond indenter (THV-6) was used for variable cutting depth testing, as shown in Figure 3. The diagonal and relative angles of the indenter are 136°. The Vickers diamond indenter was installed on the HSK63 shank, as shown in Figure 4. The test material was a 2D-SiCf/SiC ceramic composite, and the workpiece size was 15 × 12 × 3 mm. Before the test, the material only needed simple treatment. The test material was fixed on the machine tool with a particular fixture, and the machine tool program was controlled to maintain a certain angle. Thus, the experimental results were obtained by cutting the indenter on the surface of the material. Then, the plate was screwed firmly onto the K9527B dynamometer (Kistler, Winterthur, Switzerland). Finally, the dynamometer was fixed on the machine tool table with a fixture, as shown in Figure 4. The surface features were measured using a laser confocal microscope (LEXT OLS3000, OCPNY, Tokyo, Japan) to obtain the roughness.

The microhardness was tested with a CLEMEX ST-2000 (Clemex, Technologies, Montreal, QC, Canada) digital display. The test instrument can realize the automatic loading and unloading process, as it is equipped with an automated electric loading platform. The working accuracy of the *X* and *Y*-axis can reach 0.5 μm, while the autofocus function can reach 0.1 μm on the *Z*-axis. The effects of three directions on the surface quality and cutting force of SiCf/SiC ceramic composites with the same amplitude were studied. The workpiece is clamped in a triaxial piezoelectric dynamometer (Kistler 9257B, Kistler, Winterthur, Switzerland) with a sampling frequency of 7 kHz. On the surface of the workpiece, the two components of the experimental scratching forces (SFs) are *F_x_* and *F_z_*. Where *F_n_* = −*F_z_* (normal SF) and *F_t_* = −*F_x_* (tangential SF). Where *S_v_* is the scratching feed speed, and *S_d_* is the scratching depth, as shown in Figure 5a. By rotating the table of the DMG machining center around the axis by a certain angle, the testing of the variable cutting depth can be accomplished, as shown in Figure 5b. Then, the conventional scratching and ultrasonic-assisted scratching were observed using scanning electron microscopy (SEM).

### 2.3. Scratching Experiment

Due to the anisotropy and heterogeneity of the material, the material can be divided into three typical planes, S1, S2, and S3, as shown in Figure 6. According to the designed direction and selected cutting surface in the figure, S1 represents the vertical plane composed of 0° fiber bundles and 90° fiber bundles, S2 is composed of cross-fibers and 0° fiber bundles, and S3 is mainly composed of 90° fiber bundles and cross-fibers. Subsequently, ultrasonic scratching tests were carried out on the three vertical surfaces, S1, S2, and S3, respectively. The parameters used in the scratching process are shown in Table 2. For each group of test data, the scratching mode, as shown in Figure 7, was adopted for these three surfaces, and each group of test data was collected five times; afterwards, the mean scratching force was calculated. After the test, each workpiece was placed under the scanning electron microscope (SEM) to observe the specific situation after scratching. Finally, the roughness was measured using a laser confocal microscope (LEXT OLS3000) instrument.

## 3. Results and Discussion

### 3.1. Scratching Force and Surface Morphology of Three Typical Braided Surfaces

In the SEM images of the two kinds of machined microsurface morphology in Figure 8, it can be seen that the morphology of the fiber fracture surface is irregular, and the ultrasonic vibration-assisted scratching (UVAS) method is slightly better than the conventional scratching (CS) method. The removal patterns of composites processed in both ways are similar. The overall performance of the fiber is one of brittle fracture removal, mainly in the form of collective crushing, fiber fracture, the fiber–matrix interface being off-site, peeling, and so on. The strength of the matrix is greater than that of the fiber, and the interface adhesion between the fiber and the matrix is relatively weak, meaning that many pieces fall away. Therefore, the mode of the SiCf/SiC removal is brittle.

The force data of S1, S2, and S3 on three surfaces were collected by a 9257B dynamometer, and ultrasonic-assisted machining was carried out using a DMG-ultrasonic70-5-linear machining tool. It can be seen from the results that the scratching force Fz on the three surfaces and the plane forces *F_x_* and *F_y_* change periodically and uniformly. *F_x_* and *F_y_* constitute the tangential force, and the normal scratching force is larger than the tangential forces. The weaving form of the surface of the material changes periodically, and the corresponding scratching force is also different; the surface scratching force is measured by averaging the scratching force when calculating the scratching force. Changes in the scratching speed and depth influence the SF, as shown in Figure 9 and Figure 10, respectively.

It can be seen from Figure 9a,b that with the increase in velocity, the scratching force on each scratching surface increases significantly and obviously; this may be because the selected velocity values vary wildly. With the increase in scratching depth from 20 μm to 50 μm, the scratch force on the three surfaces increases evenly, but not significantly. An increase in scratch depth directly leads to an improvement in the material removal rate. However, from Figure 9 and Figure 10, it is clear that the surface scratching force of S3 is greater than the other two surface forces. Under the influence of the anisotropy of the woven and laminated structures of the composite, the normal and tangential scratching forces of the different plane engravings follow the size order S3 > S2 > S1. As shown in Figure 9, the tangential scratching force (*SF_t_*) of the three planes is less than the normal scratching force (*SF_n_*), which is about half of *SF_n_*; with the increase in feed speed, both *SF_n_* and *SF_t_* increase. Because the single particle increases in unit time with the increase in feed speed, the processing efficiency is improved, and the amount of removal achieved per unit of time is increased; therefore, the SFs increases. In Figure 10, it is shown that the SFs increase with increasing depth. The *SF_t_* and *SF_n_* in the S3 plane change significantly more than the forces in the two planes of S1 and S2, and these differences are mainly attributed to the lamination factor. In the S3 plane, the fibers are characterized by *X*-*Y* fiber weaving; they are then hot-melt laminated in the *Z* direction to form a composite material. The two-dimensional structural properties significantly reduce anisotropy, so the shear strength and tensile strength in the lamination plane are the worst combination therein [32]. At the same time, the debonding and fiber peeling between the layers are very tolerant of the fiber/matrix of the plane, so the scratching force of the laminate plane is the lowest.

Two scratching methods were used according to the scratching direction shown in Figure 11. According to the DMG operation manual and display interface, the accuracy of the rotary axis is 0.001°. The turntable was rotated by 0.115° to achieve a 1 mm scratching at a depth of 2 μm. In the beginning, when the blade tip is drawn, the depth of the cutting is shallow, the damage to the surface of the composite material is minimal, and the difference in the comparative cutting force is not apparent. However, it can be seen that the primary forms of failure are matrix fracturing, fiber debonding, peeling, and fiber fracturing. The ceramic fiber material S1 has a transverse and longitudinal fiber bundle braided layer on its surface. Therefore, parallel and identical woven forms were chosen before scratching observations began.

When the indenter is used for normal scratching, the matrix is slightly deformed in the initial stage, after which many long fibers come off and the matrix is seriously fractured. The microscopic surface morphology is shown in Figure 12a. When the material is scratched using ultrasonic vibration, there is no peeling phenomenon over a long distance, and the roughness of the matrix is smaller than that achieved with ordinary crushing, as can be seen in Figure 12b. The fiber’s breaking time is short, resulting in a low fiber debonding rate and a smooth break removal.

According to the analysis in Figure 12 and Figure 13, the brittleness of the composite matrix is greater than that of the SiC fiber in the CS process, and the SiC matrix begins to crack due to scratching friction. The fracture energy is dispersed as the crack extends to the interface between the matrix and fiber. The fracture is temporarily blocked and begins to expand at the boundary with the increase in shear and extrusion pressures. In the CS process, only SiC fibers are crushed into fragments, and longer fibers may be separated from the matrix. In the process of UVAS, the axial impact is added based on CS. The *Z*-axis is attached, and ultrasonic vibration can cut the fiber into smaller pieces that are easier to remove. As the tool moves up and down, the tool cools effectively, so the tip holds well. At the same time, the up and down vibration causes the slag and fibers to excrete more easily; thus, the surface quality improves.

### 3.2. Ultrasonic Scratch Morphology Observation and Removal Mechanism Analysis

SiCf/SiC composites are composed of the SiC matrix and SiC fiber-braiding stack inside the complex; the material’s removal occurs in two stages. The first is when the brittle matrix failure occurs. The internal SiC fiber is debonded and stripped; a schematic diagram of this is shown in Figure 14. When the diamond touches the material surface, plastic deformation occurs due to stress. With the increase in the diamond’s entry depth, the stress on the surface gradually increases, and the surface begins to feature brittle fractures and internal, transverse, and longitudinal radial cracks [44,45].

The SiCf/SiC composite comprises a SiC matrix phase and SiC fiber phase. The removal mechanism of the composite is different from that of ordinary materials. The SiCf/SiC composite features the toughening effect of SiC fiber, in addition to the brittleness and fracturability of the matrix. A single fracture diagram of the SiCf/SiC composite is shown in Figure 14 below. When the abrasive particles are in contact with the surface of the SiCf/SiC composite, the material first produces a small deformation due to extrusion and shear stress. With the increase in abrasive particle pressure, the cutting depth also increases, and the surface stress of the abrasive particles on the SiCf/SiC material increases. When the stress increases to a particular value, the material begins to show transverse cracks and longitudinal radial cracks. With the continuous action of the abrasive particles, the cracks in both directions extend further, and the degree of damage is related to the material’s fracture toughness. When the stress of abrasive particles on the material is less than the fracture toughness, the longitudinal radial cracks’ propagation direction changes and deflects in the matrix, as shown in Figure 14a. Suppose that the stress of the abrasive particles on the material continues to increase and is greater than the fracture toughness; in this case, the crack will expand to the SiC fiber, and the fiber will suffer from brittle fracture, peeling off from the matrix, and other forms of damage, as shown in Figure 14b.

According to the above experimental data, each surface will undergo the following stages: (1) When the extrusion pressure is small, the surface undergoes slight plastic deformation. (2) Due to the brittleness of the SiC matrix being greater than that of the SiC fiber, the SiC composite cracks and the stress increases. (3) When the pressure increases, the interface cracks between fiber and matrix continue to expand, and the phenomena of fiber and matrix peeling, shedding, and tearing occur. (4) The silicon carbide fiber breaks directly into pieces and falls from the material surface.

In Figure 15, it can be observed that ultrasonic-assisted vibration grinding can scratch the SiC fiber in different directions, and the results are entirely different. In Figure 15a,b, in the horizontal transverse direction, we can see that the average score of the fibers due to tangential force is far more considerable than the SiC fiber and substrate, hence the long fiber breaks off after ultrasonic vibration machining is added; this is due to the up and down shaking force, which cuts into the direction of short fibers, therefore producing a small loss. In Figure 15c,d, with horizontal tangential scratching, the fibers are fractured due to shear and extrusion, and the fibers on both sides of the fracture spill off on to the matrix, but are relatively uneven. However, the ultrasonic effect means that the press head used for processing on both sides of the crack can cause the fiber to be more evenly cut. As shown in Figure 15e,f, there are many fiber holes in SiCf/SiC ceramic composites, and the fiber fracture toughness cannot withstand the large cracks caused by axial force after the ordinary cutting of the SiC matrix. However, when the ultrasonic-assisted fraction vibrates upward and downward, the matrix can form small fault blocks, and the chips can discharge in time without secondary damage. The fiber toughness can retain a small matrix block on the workpiece without causing a large area of surface breakage, thereby improving the surface roughness.

### 3.3. Study on the Influence of Ultrasonic Vibration on Surface Roughness

The difference in roughness cannot be fully seen simply by viewing the 2D morphology. The 3D characterization method can illustrate the difference between the height of each position and can accurately and comprehensively reflect the different microstructures. By establishing the function relation, the average arithmetic height *S_a_* was calculated using a computer. *S_z_* expressed the surface topography of the composite, and the surface roughness was objectively expressed as *S_q_*, where *S_a_* represents the mean value of an absolute value within a limited region.
(1)$$S_{a}=\frac{1}{MN}\sum_{j=1}^{N}\sum_{i=1}^{M}|\eta \left(xi,yj\right)|$$

*x_i_* and *y_j_* are the distances from the *x*-direction and *y*-direction points on the outline to the reference line, respectively. *M* and *N*, respectively, are included in order to adopt the *x*-direction and *y*-direction measurement points in the region; *η* is the measurement accuracy coefficient.

The *S_q_* (root mean square height) is equal to the standard deviation of the height distribution, and the mean square value is also known as the validity in physics, which is a convenient statistical method.
(2)$$S_{q}=\frac{1}{MN}\sqrt{\sum_{j=1}^{N}\sum_{i=1}^{M}|\eta^{2}\left(xi,yj\right)|}$$

Figure 16a shows that the tangential force is larger than the scratching force as a whole; both decrease first and then increase with the increase in amplitude. When the amplitude is 2.5 μm, the normal and tangential forces are the smallest. The white light interferometer was used to measure the sampling surface of each scratch; the obtained data were statistically averaged and plotted into a chart, as shown in Figure 16b, for a comparative analysis of roughness. It can be seen that the roughness of the ultrasonic vibration-scratched surface is significantly greater than that of the ordinary scratched surface, and the *S_q_* roughness is increased by 34~51%.

In the process of face grinding, due to ultrasonic vibration, the diamond material’s abrasive grains maintain discontinuous contact with the workpiece material and impact the workpiece surface at a high frequency. Since the speed of impact is extremely fast and much greater than the speed of feeding, the high-frequency result is the primary removal mode in the face grinding process. The ratio of normal grinding force to tangential grinding force (*F_n_*/*F_t_*) is used to evaluate the grinding performance of the grinding wheel. The greater the grinding force, the worse the performance of the grinding wheel. As shown in Figure 17b, the quality is worst with a surface grinding amplitude of 0 μm. As the amplitude increases, the surface quality deteriorates, a trend similar to the grinding force ratio presented in Figure 17a. Notably, when the amplitude is 1.25 μm, the surface quality is the best, and the grinding force ratio is the smallest.

Through ultrasound-assisted scratching and grinding experiments, it is not difficult to see the influence that these processes have on reducing the anisotropic processing of SiCf/SiC composites to a certain extent. SiC fibers are cut into multi-truncated fibers by ultrasonic impact and are removed by short fibers, which reduces the breaking and peeling of SiC fibers in conventional scratching. With the amplitude increase in the marking experiment, the surface scratching force first increased and then decreased. The roughness measurements gradually increased, but all were lower than those of ordinary machined surfaces. The end face grinding experiment further verifies that ultrasonic amplitude has a significant influence on the improvement of surface roughness. Said ultrasonic amplitude reduces the contact time between the abrasive particles and the material; thus, the grinding force is significantly reduced, and the chips can be discharged in time. The ultrasound-assisted impact promotes some small cracks in the composite, making the material easy to remove. Both experiments verify that the surface roughness and quality of composite materials are improved by ultrasonic-assisted processing.

## 4. Conclusions

This work mainly studies the grinding mechanism of 2D-SiCf/SiC using ultrasonic vibration grinding. The characteristics of scratching force and the surface morphology of three typical braided surfaces were studied. The following conclusions were obtained:

Observing the contrast between ultrasonic scratching and ordinary scratching, conventional scratching prompts severe fiber peeling and debonding, as well as larger matrix fracture cracks. In ultrasonic vibration mode, the fracture fiber becomes shorter, the chip becomes smaller, the crack fracture becomes smaller, and the overall surface smoothness is relatively better.

When three planes are scratched using ultrasonic vibration, the changing trend in scratching force is roughly the same. With an increase in the scratching speed, the normal and tangential scratching forces also increase. With an increase in scratching depth, the normal and tangential scratching forces show little change. However, with a change in the direction of the indentation head as it contacts the fiber braid, the scratching force is significantly different. The scratching force can be divided into S1, S2, and S3. The scratching forces on the three surfaces generally follow the order S3 > S2 > S1.

With the same feed rate and grinding depth, different amplitudes can reduce the grinding interaction time between abrasive particles and materials. Their combined action reduces the force and signs of subsurface damage such as matrix cracks and fiber debonding. The surface fibers of the material are subject to amplitude vibration; truncated short fibers also become neater and chips are easily discharged, so this form of processing is more conducive to a high-quality material surface.

## Figures and Tables

**Figure 1 micromachines-14-01350-f001:**
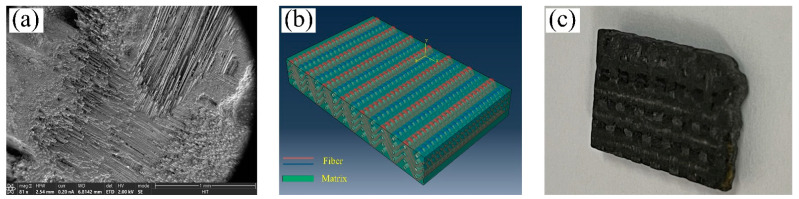
Schematics of the structure of the SiCf/SiC workpiece material: (**a**) SEM image, (**b**) plane woven structure of the SiC fibers and matrix, and (**c**) images of the SiCf/SiC composite.

**Figure 2 micromachines-14-01350-f002:**
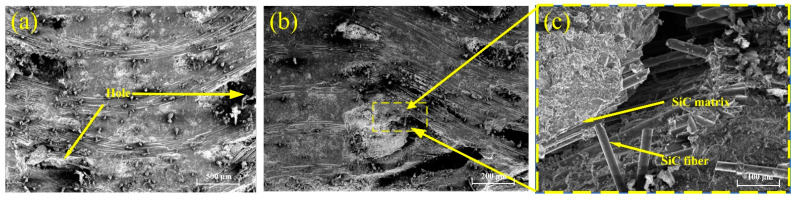
SiCf/SiC composite structure: (**a**) SiC fiber braided mesh layer, (**b**) section diagram of SiCf/SiC composites, and (**c**) ceramic matrix.

**Figure 3 micromachines-14-01350-f003:**
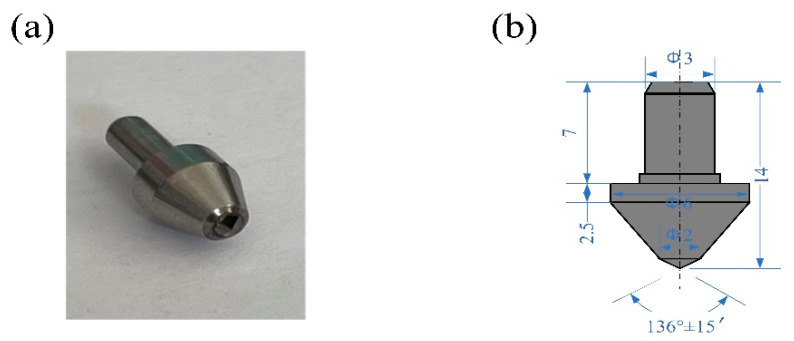
Pictures and detailed parameters of the Vickers diamond indenter. (**a**) Physical image of indenter, and local enlarged image; and (**b**) experimental indenter size parameters.

**Figure 4 micromachines-14-01350-f004:**
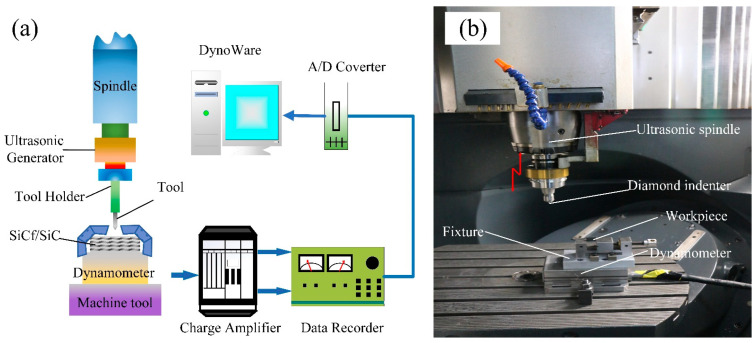
System set-up for scratching tests: (**a**) schematic diagram of the overall structure; and (**b**) photo of the experimental device.

**Figure 5 micromachines-14-01350-f005:**
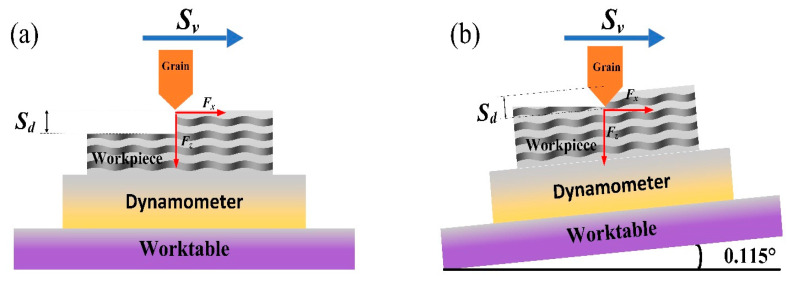
Single-grain scratching experiment. (**a**) single grain conventional scratching and (**b**) single grain scratching with a gradual depth.

**Figure 6 micromachines-14-01350-f006:**
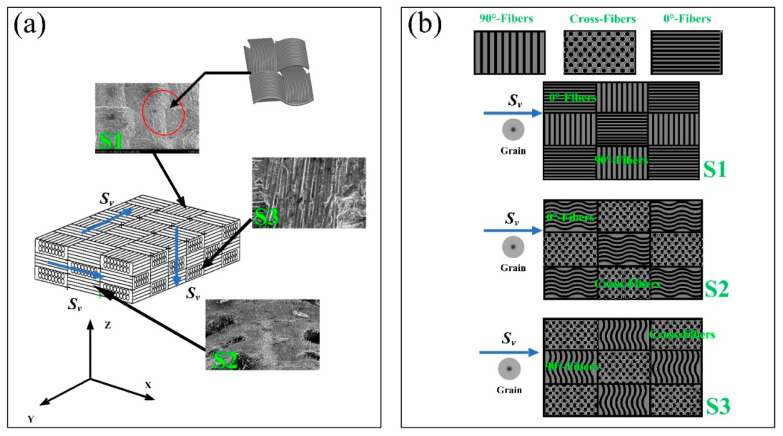
Definition of typical surfaces and scratching directions of composites: (**a**) 3D braided structure and cutting directions of SiCf/SiC materials; and (**b**) definitions of the scratched surfaces of S1, S2, and S3.

**Figure 7 micromachines-14-01350-f007:**
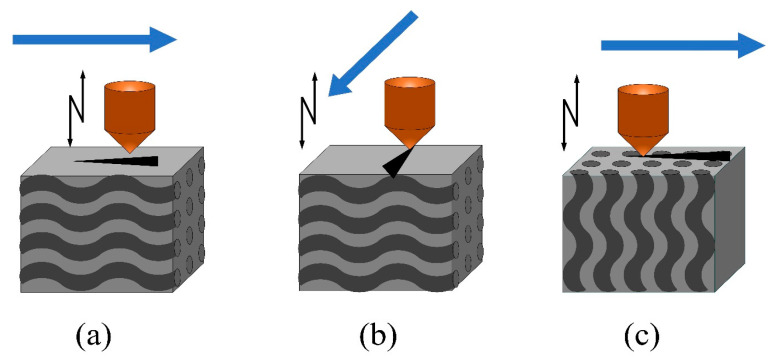
Definition of the scratching mode and the given direction of vibration: (**a**) longitudinal scratching; (**b**) transverse scratching; and (**c**) normal scratching.

**Figure 8 micromachines-14-01350-f008:**
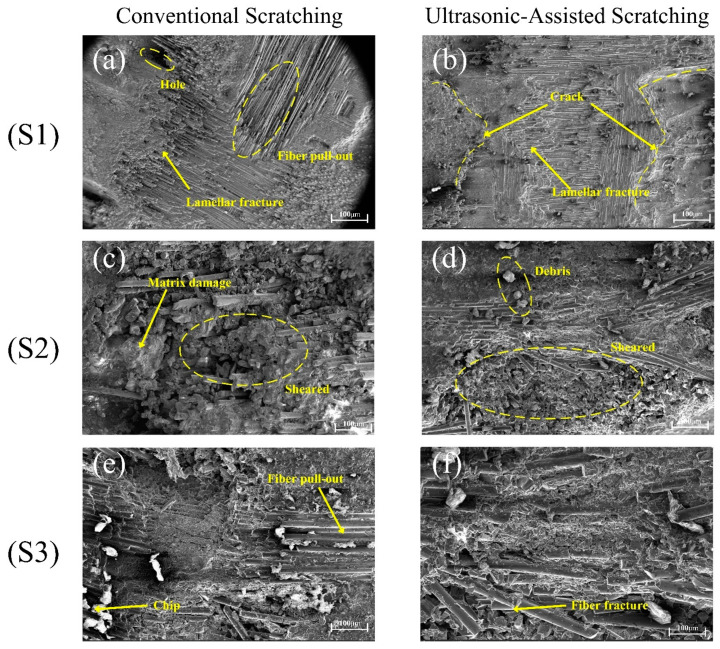
Morphologies of the scratched surfaces of S1 (**a**,**b**), S2 (**c**,**d**), and S3 (**e**,**f**).

**Figure 9 micromachines-14-01350-f009:**
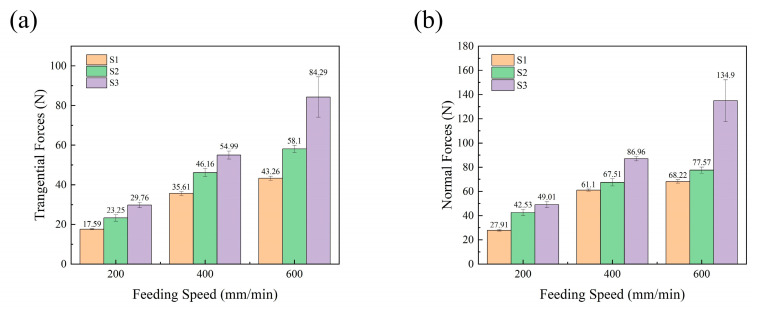
The effects of scratching speed on components of SF, under ultrasonic vibration machining, (*A_m_* = 3 μm, *S_d_* = 20 μm): (**a**) tangential SF and (**b**) normal SF.

**Figure 10 micromachines-14-01350-f010:**
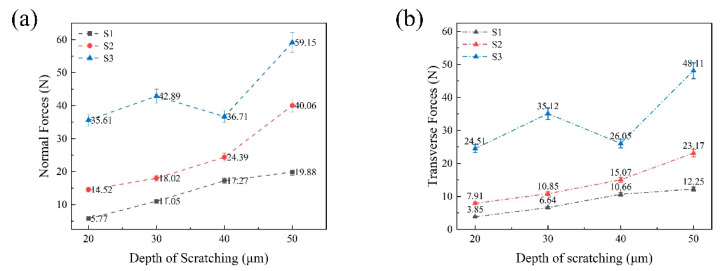
The effects of scratching depth on components of SF under ultrasonic vibration machining, (*A_m_* = 3 μm, *S_v_* = 200 mm/min): (**a**) normal SF, and (**b**) tangential SF.

**Figure 11 micromachines-14-01350-f011:**
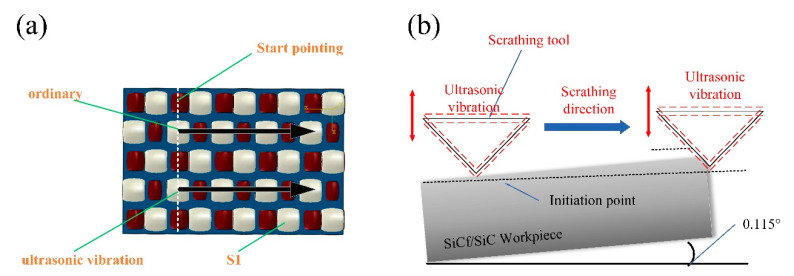
Diagram of fiber and scratch directions: (**a**) is the top view and (**b**) is the right view.

**Figure 12 micromachines-14-01350-f012:**
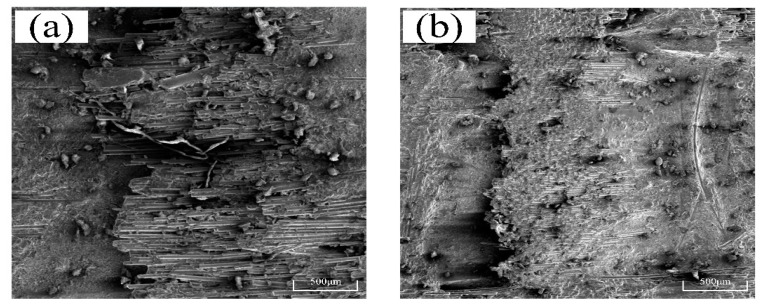
Scanning electron morphologies of the scratched surface of S1, *S_v_
*= 10 mm/s, *θ* = 0.115°: (**a**) *A_m_
*= 0 μm, and (**b**) *A_m_
*= 3 μm (*A_m_* represents amplitude).

**Figure 13 micromachines-14-01350-f013:**
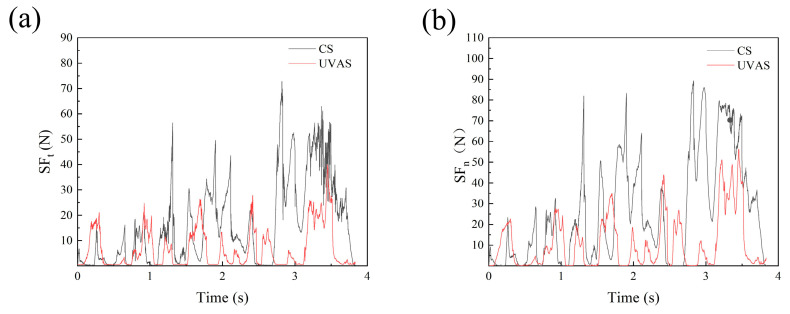
Variation in the measured scratching force (SF) with the cutting depth on the surface of S1: (**a**) conventional scratching and (**b**) ultrasonic vibration-assisted scratching.

**Figure 14 micromachines-14-01350-f014:**
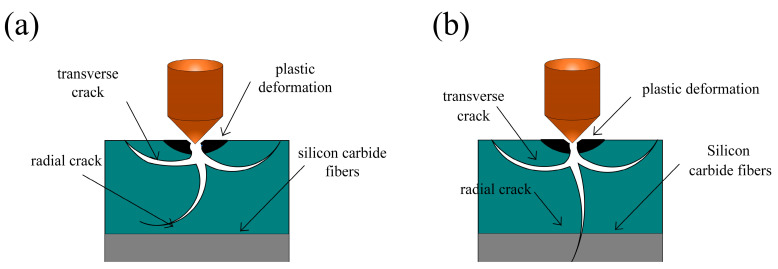
Single-particle grinding fracture model of SiCf/SiC composites: (**a**) crack deflection and (**b**) fiber breakage.

**Figure 15 micromachines-14-01350-f015:**
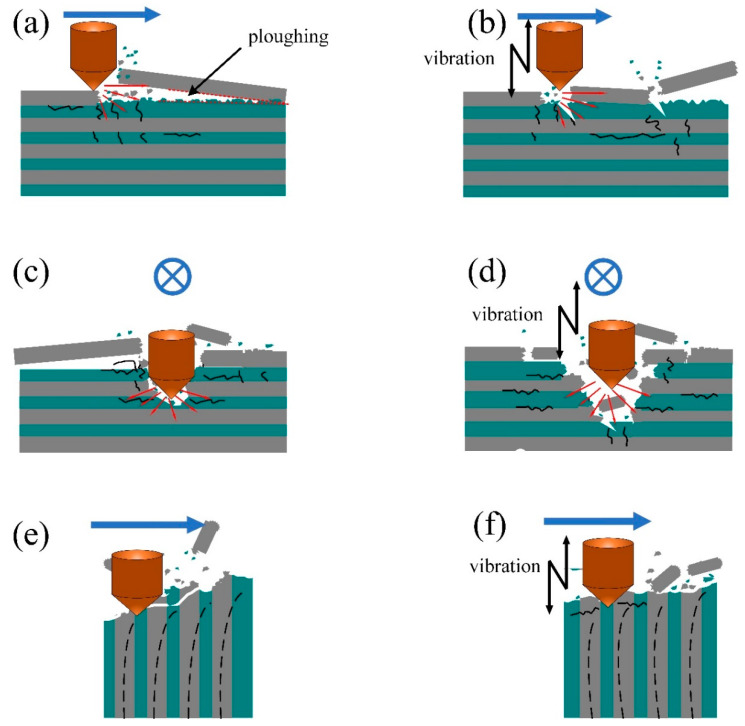
Influence of ultrasonic vibration on material removal mode: (**a**,**c**,**e**) represent ordinary scratching; (**b**,**d**,**f**) represent ultrasonic vibration-assisted scratching.

**Figure 16 micromachines-14-01350-f016:**
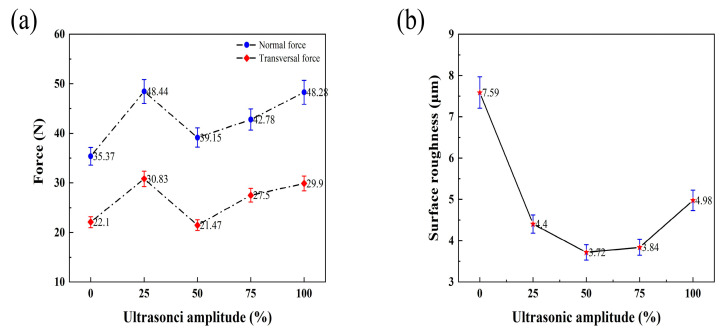
Variation in surface scratching force and surface roughness under different amplitudes (*S_v_* = 200 mm/min, *A_p_* = 20 µm, and *A_m_
*= 5 μm). (**a**) The tangential force is larger than the scratching force as a whole; both decrease first and then increase with the increase in amplitude. (**b**) The white light interferometer was used to measure the sampling surface of each scratch.

**Figure 17 micromachines-14-01350-f017:**
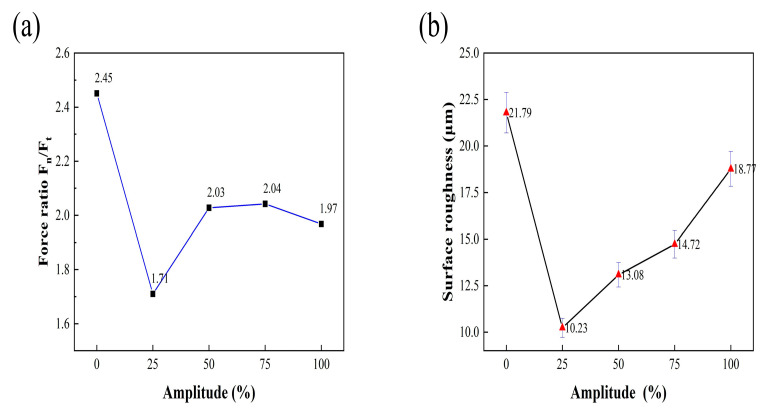
Effect of ultrasonic amplitude on grinding force ratio and surface roughness (*S* = 12,000 r/min, *S_v_
*= 25 mm/min, *A_p_
*= 25 μm, and *A_m_
*= 5 μm). (**a**) The grinding force ratio, The greater the grinding force, the worse the performance of the grinding wheel. (**b**) The ratio of normal grinding force to tangential grinding force (Fn/Ft) is used to evalu-ate the grinding performance of the grinding wheel.

**Table 1 micromachines-14-01350-t001:** Material properties of SiCf/SiC [43].

Parameters	Density (g/cm^3^)	Fiber Volume Fraction (%)	E-Modulus (GPa)	Tensile Strength (Mpa)	Flexural Strength (Mpa)	Compression Strength (Mpa)
Values	2.5 ± 0.03	23 ± 3	220 ± 10	271.4 ± 29.0	500 ± 50	454.0 ± 18.2

**Table 2 micromachines-14-01350-t002:** Scratching parameters for single-grain scratching dates.

Surfaces	Parameters	Values	Ultrasonic Amplitude (μm)
S1, S2, S3	Feeding speed *S_v_* (mm/min)	200, 400, 600	3
	Scratching depth *S_d_* (μm)	20, 30, 40, 50	

## Data Availability

Data available on request from the authors.

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
