# Peer review of "Investigation of Cutting Force and the Material Removal Mechanism in the Ultrasonic Vibration-Assisted Scratching of 2D-SiCf/SiC Composites"

_micromachines, 2023, doi:10.3390/mi14071350_

Round 1
Reviewer 1 Report
The present work entitled “Investigation of cutting force and the material removal mechanism in the ultrasonic vibration-assisted scratching of 2DSiCf/SiC composites” investigates cutting force and the material removal mechanisms in the ultrasonic vibration-assisted scratching of 2D SiCf/SiC composites via single-grain scratching experiments. The paper is well framed and written. Some comments to further improve the quality of this paper are given below:
1. The sources of the properties listed in Table 1 should be given. If they are measured by the authors, the method should be clarified. Otherwise, the references should be cited.
2. The surface state of the samples before scratching should be measured and given in the manuscript.
3. The meanings of different colors in Figs. 1 (b) and (c) should be explained.
4. The manufacturer, model and country of the instruments and equipment used in this study should be given in the manuscript in such a format: equipment name (model, manufacturer, country).
The quality and completeness of the manuscript are good. It can be accepted for publication after minor revisions.
Please carefully read and revise English writing of the manuscript.
Author Response
Response to reviewer#1”s comments
Point 1: The sources of the properties listed in Table 1 should be given. If they are measured by the authors, the method should be clarified. Otherwise, the references should be cited.
Response: Thank you for your comment. The parameters in Table 1 are derived from the literature " Effect of fabric structure on the permeability and regeneration ability of porous SiCf/SiC composite prepared by CVI " reference, which I have added to the back of Table 1.
Point 2: The surface state of the samples before scratching should be measured and given in the manuscript.
Response: Thank you for your comment. Indeed, Since there are too many samples to show all of them, I have selected a clear sample photo to be placed in figure 1(c) to show the solid structure of the surface.
Point 3: The meanings of different colors in Figs. 1 (b) and (c) should be explained.
Response: Thank you for your suggestion. I have modified your suggestion and added graphic annotations in the figure to explain the meaning of each color.
Point 4: The manufacturer, model and country of the instruments and equipment used in this study should be given in the manuscript in such a format: equipment name (model, manufacturer, country).
Response: Thank you for your comment and we agree with you very much. I have changed the original article sentence 'DMG-ultrasonic 70-5-linear machining tool' to 'CNC five-axis machining center as you said ( Ultrasonic 70-5 linear, DMG MORI, Germany).'
Reviewer 2 Report
Manuscript ID: micromachines-2473915
Dear Editor,
The authors discussed the effect of feed speed, depth, and ultrasonic amplitude on the scratching force and surface. They prepared three surfaces of SiCf/SiC ceramic materials.
1- It is better to define SiCf as SiC fibre from the beginning of the research to avoid misuderstanding.
2-The start of the introduction section is great, however, no any references were cited.
3-Use first Capitalize letter to define any term, for example, Electrical Discharge Machining (EDM), .......etc. The same procedure for other terms used in the current research.
4-Table 1, for ceramic materials, flexural strength should replace the bending strength term.
5-density is in g/cm3
6-certain angle in page 5
7-page 12, SiC replace sic !
8-Figure 16 c or Figure 16 b ?
9-Figure 16 c is not shown.
10-The experimental results show that the ultrasonic vibration scratching force is significantly lower than normal.
This sentence in the Abstract section is not clear.
Simple punctuation and grammatical correction helps the improvement of the submitted article
Author Response
Response to reviewer#2”s comments
Point 1: It is better to define SiCf as SiC fibre from the beginning of the research to avoid misuderstanding.
Response: Thank you for your comment, and this comment is very important to us. I have added a note after the sentence ‘silicon carbide fiber-reinforced ceramic matrix composites” in page 1 paragraph 1, so that it becomes" silicon carbide fiber-reinforced ceramic matrix composites (SiCf/SiC) ".
Point 2:. The start of the introduction section is great, however, no any references were cited.
Response: Thank you for your suggestion, and we agree with you very much. I realize that there is indeed less literature cited earlier, and I have looked up some relevant literature and added it in the right places.
Point 3: Use first Capitalize letter to define any term, for example, Electrical Discharge Machining (EDM), .......etc. The same procedure for other terms used in the current research.
Response: Thank you for your suggestion. This was a grammatical oversight on my part, and I have changed the initial letters of the terms defined in the text to be capitalized. I hope that all corrections are satisfactory.
Point 4: Table 1, for ceramic materials, flexural strength should replace the bending strength term.
Response: Thank you for your suggestion. I have changed "bending strength" to "flexural strength" as you requested.
Point 5: density is in g/cm3.
Response: Thank you for your comment. I have corrected the errors in the article form as you suggested. Table 1 Change " (g/cm3) " from " (g/cm3) " .
Point 6: certain angle in page 5.
Response: Thank you for your affirmation of our use of English. And we have carefully proofread the semantics mistakes of this paper.
Point 7: page 12, SiC replace sic !
Response: Thank you for your suggestion. I have corrected all the mistakes in English.
Point 8: Figure 16 c or Figure 16 b ?
Response: Thank you for your suggestion. And we have carefully proofread the semantics mistakes of this paper.
Point 9: Figure 16 c is not shown.
Response: Thanks for your careful correction. The original manuscript had Figure 16(c), which was later removed because it was not very useful. My negligence did not talk about the manuscript carefully proofread, and now I have corrected it to remove its ''Fig16 (c)''.
Point 10: The experimental results show that the ultrasonic vibration scratching force is significantly lower than normal.
Response: Thank you for your suggestion. You are absolutely right. From the figure 17 , it can only be seen that the ultrasonic assisted scratching is slightly less than the conventional scratching force. I have changed the original sentence to " the ultrasonic vibration assisted scratching force is slightly lower than the conventional scratching force".
Reviewer 3 Report
The authors performed a series of experiments comparing ordinary scratching processes with ultrasonic-assisted scratching for SiCf/SiC composites. They claim that ultrasonic scratching provides higher quality finishing of the surfaces. The work is overall well done and well presented, but there is the need of some minor revisions as the data presented doesn't sufficiently support their main claim.
In particular, the imply that higher quality in finishing is given by lower surface roughness and at the end of section 3 they say "The roughness measurements gradually increased, but all were lower than those of ordinary machined surfaces". Yet, no data for the ordinary machined surfaces is shown so the authors cannot make this conclusion. Same for the sentence "Both experiments verify that the surface roughness and quality of composite materials are improved by ultrasonic-assisted processing." and similar sentences in the abstract and throughout the manuscript.
The other main concern that I have is about the number of experiments run to make the conclusion statistically significant. How do the authors support this?
The final main concern that I have regards the whole discussion in section 3 with the schemes of Figs 14 and 15. It would benefit a lot from having references to previous figures of post-experiment pictures to show that the described mechanism is supported by experimental evidence. If this is not possible, then the whole discussion in section 3 loses significance.
Further minor comments:
In the abstract: unclear the link between S1-S2-S3, so S3>S2>S1 does not make sense until much later in the manuscript. Remember that the abstract should be understandable on its own.
"NASA [...] identified SiCf/Sic..." -> need reference
"EDM method": need to use the actual words before introducing acronym
Some citations (Liu et al, Yang et al, Ning et al) have the reference number at the end of the sentence instead of after the authors' name(s).
"Surface roughness reduced by 30%" - what kind of roughness? RMS? Hurst exponent?
Figure 3a: picture is not bright enough, no object in the picture to understand scale. Personally I don't think the picture is needed.
Page 5, definition of SF: "positive pressure" is ill defined - a 'pressure' has the units of a force/area but it seems that Fn = -Fz is actually a force, and 'positive' has no meaning given the no frame of reference is provided
"where Sv is the scratching feed speed, and Sd..." why 'where'? They were never introduced before.
"were carried out on the S1 surface, respectively" - why 'respectively'? Sounds irrelevant as a word.
S1, S2, S3 are used both for sample names and for typical planes of the material. The two should be better distinguished.
What is the scribe force? And the scoring force? And the marking force? What is SFn? How does it differ from Fn? Same for SFt and Ft. Also, using 'normal' for a tangential component of the force is confusing, I recommend to find another term.
Figure 10 (and others): numbers on markers are not readable.
Why is mu>0.7,0.8 in many cases and never mu<0.5 (Figs. 9 and 10), when in the introduction the material is described a low friction one?
First paragraph in Section 3.2 describing plastic contact and radial crack: cite relevant literature (e.g. Lawn, Fracture of brittle solids) as it is a well known phenomenon.
Simple punctuation and grammatical correction helps the improvement of the submitted article
Author Response
Response to reviewer#3”s comments
Point : Yet, no data for the ordinary machined surfaces is shown so the authors cannot make this conclusion.
The other main concern that I have is about the number of experiments run to make the conclusion statistically significant. How do the authors support this?
The final main concern that I have regards the whole discussion in section 3 with the schemes of Figs 14 and 15. It would benefit a lot from having references to previous figures of post-experiment pictures to show that the described mechanism is supported by experimental evidence. If this is not possible, then the whole discussion in section 3 loses significance.
Response: Thank you very much for your comments and professional advice. These points contribute to the academic rigor of our articles. According to your suggestions and requests, we have made corrections to the revised manuscript and answered your questions.
Ultrasonic vibration grinding experiments were done later, and different amplitude comparison experiments were used, where Figure 17 shows that there is 0 μm amplitude representing ordinary grinding processing. This can indicate that the surface roughness of ordinary machining is inferior to that of ultrasonic vibration grinding.
For your question about the number of experiments I may not have done a perfect job to conduct multiple sets of experiments for repeated verification. Because of the expensive material costs, the number of processing times was limited and only a minimum of comprehensive experiments were conducted at first. However, I will continue to conduct angular and grinding experiments to study ultrasonic vibration to present all these theoretical bases.
The main content of the third section is to summarize the schematic diagram of the surface force principle through the literature study of ceramic matrix composites and related fiber structural property characteristics, and combined with the surface characteristics and force characteristics imaged by the microscope made in this paper. It is possible that the basis is not perfect now, and subsequently I will use transmission microscopy electron micrographs to observe the specific differences between the two to confirm.
Point 1: In the abstract: unclear the link between S1-S2-S3, so S3>S2>S1 does not make sense until much later in the manuscript. Remember that the abstract should be understandable on its own.
Response: Thank you for your comment. We have revised the abstract in accordance with your request.
Abstract— Ultrasonic assisted grinding (UAG) has been widely used in the manufacture of hard and brittle materials. However, the process removal mechanism has not been elucidated and its potential has not been fully exploited. In this paper, the mechanism of material removal is analyzed by ultrasonic assisted scratching. Three distinct surfaces (S1, S2, S3) were selected on the basis of the braided and laminated structure of fiber bundles. The directions are classified as longitudinal (∥), transverse (⊥) and normal (⊙) according to the three modes of scratching defined by the direction of grain motion with respect to the fiber axis. Where S1 plane is (∥) and (⊥); S2 plane is (∥) and (⊙); S3 plane is (∥) and (⊥).The ultrasonic-assisted scratching experiment is carried out under different conditions, and the scratching force (SF) of the tested surface will fluctuate periodically. the conditions of different feed speeds, depths, and ultrasonic amplitudes, the normal scratching force (SFn) is greater than the tangential scratching force (SFt), and the average scratching force on the three surfaces is generally S3 > S2 >S1. Among the three processing parameters, the speed has the most significant influence on thescratching force, while the scratching depth has little influence on the scratching force. Under the same conditions and surface cutting mode, the ultrasonic vibration assisted scratching force is slightly lower than the conventional scratching force. The scratching force decreases first and then increases with the amplitude of ultrasonic vibration. Because the fiber undergoes a brittle fracture in the ultrasonic-assisted scratching process, the matrix is torn, and the surface residues are discharged in time; therefore, the surface roughness is improved.
Point 2: "NASA [...] identified SiCf/Sic..." -> need reference "EDM method": need to use the actual words before introducing acronym
Response: According to your nice suggestions, we have made extensive corrections to our previous draft, the detailed According to your nice suggestions, we have made extensive corrections to our previous draft, the detailed:
the National Aeronautics and Space Administration (NASA)、the high-speed civil transport (HSCT).
Point 3: Some citations (Liu et al, Yang et al, Ning et al) have the reference number at the end of the sentence instead of after the authors' name(s).
Response: We sincerely thank you for your valuable comments. We have carefully checked the literature and inserted the issue of the location of the references in the introduction of the revised manuscript. Move ''[20],[21],[41],[42]'' reference numbers to the end of the sentence.
Point 4: "Surface roughness reduced by 30%" - what kind of roughness? RMS? Hurst exponent?
Response: Thank you for your careful reading of our manuscript. The cited article focuses on an experimental study comparing ultrasonic vibration grinding and normal grinding by using a white light interferometer to measure Ra and Rz values. The resulting values were then plotted as roughness images to roughly calculate the magnitude of the surface roughness difference. Although it is not as accurate as RMS and Hurst exponent, it can show that ultrasonic vibration-assisted grinding is better than ordinary grinding.
Point 5: Figure 3a: picture is not bright enough, no object in the picture to understand scale. Personally I don't think the picture is needed.
Response: Thanks for your great suggestion on improving the accessibility of our manuscript. I have made the change you requested, replacing figure3(a) with a brighter figure. In view of the question of whether to keep the schematic, I would like to keep the picture as it is, so that I can see the shape of the tool used for the scratching experiment.
Point 6: Page 5, definition of SF: "positive pressure" is ill defined - a 'pressure' has the units of a force/area but it seems that Fn = -Fz is actually a force, and 'positive' has no meaning given the no frame of reference is provided.
Response: We sincerely thank the editor and all reviewers for their valuable feedback that we have used to improve the quality of our manuscript. I have changed the sentence ' When recording the force, the scratching force (SF) is divided into two parts: one is a positive pressure Fn= -Fz (normal force), and another is the friction force Ft, which is the sum of the force vectors in the xy plane,' to ' On the surface of the workpiece, the two components of the experimental scratching force (SFs) are fx and fz. Where Fn= -Fz (normal SF) and Ft =-Fx (tangential SF). ' in the paper
Point 7: "where Sv is the scratching feed speed, and Sd..." why 'where'? They were never introduced before.
Response: Thank you for your comment. This is my article writing is not rigorous, I did not take into account. the Sd in the paper mainly represents the scratching depth, and the labeling meaning has been explained in the original text. The latter has an Ap as the grinding depth in order to distinguish so get two symbols.
Point 8: "were carried out on the S1 surface, respectively" - why 'respectively'? Sounds irrelevant as a word.
Response: Thank you for your suggestion. As suggested by the reviewer, I have removed this sentence from the page 5 paragraph 1.
Point 9: S1, S2, S3 are used both for sample names and for typical planes of the material. The two should be better distinguished.
Response: We sincerely thank the editor and all reviewers for their valuable feedback that we have used to improve the quality of our manuscript. Since this experiment was conducted on one sample, it was divided into three planes S1, S2, S3. These three planes have different weaving structures and do not represent different samples, so I am sorry that I may not be able to classify them in further detail.
Point 10: What is the scribe force? And the scoring force? And the marking force? What is SFn? How does it differ from Fn? Same for SFt and Ft. Also, using 'normal' for a tangential component of the force is confusing, I recommend to find another term.
Response: Thank you for your decision and constructive comments on my manuscript. We have carefully considered the suggestion of Reviewer and make some changes. I redefined the ‘scratching force’, ‘tangential scratching force’, and ‘normal scratching force’ in the article for uniform symbolic marking, and corrected the centered incorrect statements to make them smooth and easy to understand.
Point 11: Figure 10 (and others): numbers on markers are not readable.
Response: We think this is an excellent suggestion. I have readjusted Figure 10 and put the font size in the figure roughly clear.
Point 12: Why is mu>0.7,0.8 in many cases and never mu<0.5 (Figs. 9 and 10), when in the introduction the material is described a low friction one?
Response: We are extremely grateful to Reviewer for pointing out this problem. I mistakenly attributed the properties of Cf/SiC composites to 2D-SiCf/SiC materials, which are denser and less flexible than C fibers, and much more difficult to prepare than Cf/SiC composites. In order to avoid the reader's understanding, I have deleted this sentence.
Point 13: First paragraph in Section 3.2 describing plastic contact and radial crack: cite relevant literature (e.g. Lawn, Fracture of brittle solids) as it is a well known phenomenon.
Response: Thank you for your comment. I have added relevant literature to supplement according to your requirement.
Reviewer 4 Report
A SiCf/SiC ultrasonic axial vibration single-particle scratching experiment was carried out in this work. The kinematics of different fiber bundles on the basis of the braided and laminated structure were studied and analyzed.. The structure of this paper is clear and reasonable, and there are sufficient experimental results. However, there are some problems that need to be focused, especially the language and formatting. Some specific concerns/comments are listed below:
1. Language expression of the abstract needs to be modified to conform to English reading habits. Please minimize the use of long sentences.
2. The format of the picture is incorrect, and they need to be adjusted as required. Figure 2 lacks a clearly visible scale; Fonts and patterns in Figure 4(a) need to be re-adjusted to make them visible; Figure 7(a), (b) and (c) have inconsistent sizes and pictures are not arranged neatly; The unit of "time(S)" in Figure 13 should be changed to "Time (s)"; In addition, the format of the graphic label "(a)(b)(c)(d)" should be consistent.
3. Parts of figures 4, 6 and 10 are blurred and need to be seen more clearly.
4. The table format of the article needs to be adjusted, and some errors exist. For example, the unit expression is incorrect: Table 1 “(g/cm3)”; Table 2 “Vf/mm/min”, “Ap/μm”;
5. The language expression in the text is not standardized enough. Such as in Figure 8, "fiber pullout" should be "fiber pullout". The sentence in Page 2 Paragraph 1 "However, with this method, the material conductivity is poor, the electrode wear is severe, and the cost is high. " has more than one subject.
6. The description “The inclination of the experimental table is 0.115° ”. it is suggested to supplement the specific measurement value, or give the working accuracy of the machine tool used.
7. What is the meaning of the symbol "Am" in this paper? Please indicate.
8. It is suggested to add pictures or tables of the corresponding relationship between ultrasonic power and ultrasonic amplitude to enhance the readability of the paper.
9. It is suggested to add experiments to verify the conclusion in the first paragraph on Page 15 " The grinding wheel generates less heat and clogging, maintaining the cutting ability of the grinding wheel's abrasive particles. " It can be analyzed through friction and wear experiment or temperature measurement.
The English Language may be improved
Author Response
Response to reviewer#4”s comments
Point 1: Language expression of the abstract needs to be modified to conform to English reading habits. Please minimize the use of long sentences.
Response: Thank you for your comment. We have revised the abstract in accordance with your request.
Abstract—Ultrasonic assisted grinding (UAG) has been widely used in the manufacture of hard and brittle materials. However, the process removal mechanism has not been elucidated and its potential has not been fully exploited. In this paper, the mechanism of material removal is analyzed by ultrasonic assisted scratching. Three distinct surfaces (S1, S2, S3) were selected on the basis of the braided and laminated structure of fiber bundles. The directions are classified as longitudinal (∥), transverse (⊥) and normal (⊙) according to the three modes of scratching defined by the direction of grain motion with respect to the fiber axis. Where S1 plane is (∥) and (⊥); S2 plane is (∥) and (⊙); S3 plane is (∥) and (⊥).The ultrasonic-assisted scratching experiment is carried out under different conditions, and the scratching force (SF) of the tested surface will fluctuate periodically. the conditions of different feed speeds, depths, and ultrasonic amplitudes, the normal scratching force (SFn) is greater than the tangential scratching force (SFt), and the average scratching force on the three surfaces is generally S3 > S2 >S1. Among the three processing parameters, the speed has the most significant influence on thescratching force, while the scratching depth has little influence on the scratching force. Under the same conditions and surface cutting mode, the ultrasonic vibration assisted scratching force is slightly lower than the conventional scratching force. The scratching force decreases first and then increases with the amplitude of ultrasonic vibration. Because the fiber undergoes a brittle fracture in the ultrasonic-assisted scratching process, the matrix is torn, and the surface residues are discharged in time; therefore, the surface roughness is improved.
Point 2: The format of the picture is incorrect, and they need to be adjusted as required. Figure 2 lacks a clearly visible scale; Fonts and patterns in Figure 4(a) need to be re-adjusted to make them visible; Figure 7(a), (b) and (c) have inconsistent sizes and pictures are not arranged neatly; The unit of "time(S)" in Figure 13 should be changed to "Time (s)"; In addition, the format of the graphic label "(a)(b)(c)(d)" should be consistent.
Response: Thank you for your comment. As you pointed out, the problems with the image format have been fixed. Scales have been added to each figure in figure 2; Fonts and patterns in Figure 4(a) was re-processed and the position was re-calibrated; redraw the sizes of figure 7(a), (b), and (c) to arrange the pictures neatly; the unit 'Time(S)' in figure 13 was changed to 'Time(s)'; the format of the graphic labels '(a) (b) (c) (d)' in the whole article is modified for uniformity, and the size and font have been consistent.
Point 3: Parts of figures 4, 6 and 10 are blurred and need to be seen more clearly.
Response: Thank you for your comment. In Figure 4, I have reassociated the parts with their corresponding notes; In Figure 6, I have enlarged and bolded all the fonts to make them legible; In Figure 10, the fonts have been enlarged and redrawn together as you suggested.
Point 4: The table format of the article needs to be adjusted, and some errors exist. For example, the unit expression is incorrect: Table 1 “(g/cm3)”; Table 2 “Vf/mm/min”, “Ap/μm”;
Response: Thank you for your comment. I have corrected the errors in the article form as you suggested. Table 1 Change " (g/cm3) " from "(g/cm3)"; Table 2 "Vf/mm/min", "Ap/μm" Change "mm/min", "μm".
Point 5: The language expression in the text is not standardized enough. Such as in Figure 8, "fiber pullout" should be "fiber pullout". The sentence in Page 2 Paragraph 1 "However, with this method, the material conductivity is poor, the electrode wear is severe, and the cost is high. " has more than one subject.
Response: Thank you for your comment. Indeed, the language expression in the paper is not standardized enough, and I have modified "pill out" to "pull-out" in Figure 8. The sentence in Page 2 Paragraph 1 "However, with this method, the material conductivity is poor, the electrode wear is severe, and the cost is high. " has been revised to " However, SiCf/SiC composites have extremely poor electrical conductivity, so the wear of tool electrode becomes lager ratio than the normal machining."
Point 6: The description “The inclination of the experimental table is 0.115° “. it is suggested to supplement the specific measurement value, or give the working accuracy of the machine tool used.
Response: Thank you for your comment. I have searched through the machine manual to find the machine working accuracy and added this sentence of "According to the DMG operation manual and display interface, the accuracy of the rotary axis is 0.001°. " to Page 11 Paragraph 1.
Point 7: What is the meaning of the symbol "Am" in this paper? Please indicate.
Response: Thank you for your comment. I have marked the meaning of am at the end of fig 12 as follows: (Am represents Amplitude ).
Point 8: It is suggested to add pictures or tables of the corresponding relationship between ultrasonic power and ultrasonic amplitude to enhance the readability of the paper.
Response: Thank you for your comment. The real ultrasonic power and ultrasonic amplitude correspondence cannot be obtained due to the DMG machine. I will focus on verifying the power and amplitude change power when I do the grinding experiment with amplitude effect next time.
Point 9: It is suggested to add experiments to verify the conclusion in the first paragraph on Page 15 " The grinding wheel generates less heat and clogging, maintaining the cutting ability of the grinding wheel's abrasive particles. " It can be analyzed through friction and wear experiment or temperature measurement.
Response: Thank you for your comment. This is my article writing is not rigorous, I did not take into account. This experiment is mainly to verify force and ultrasonic amplitude. I mentioned the problem of heat and blockage because I learned from other literature that I may not be able to conduct temperature measurement analysis and verification in such a short period of time. The solution I give is to delete this sentence to maintain the rigor of the article.